# Mild Chemical Treatment of Unsorted Urban Food Wastes

**DOI:** 10.3390/molecules28227670

**Published:** 2023-11-20

**Authors:** Elio Padoan, Enzo Montoneri, Andrea Baglieri, Matteo Francavilla, Michèle Negre

**Affiliations:** 1Dipartimento di Scienze Agrarie, Forestali e Alimentari, Università di Torino, 10095 Grugliasco, Italy; elio.padoan@unito.it (E.P.); negre.michele@gmail.com (M.N.); 2Dipartimento di Scienze delle Produzioni Agrarie e Alimentari, Università di Catania, Via S. Sofia 98, 95123 Catania, Italy; abaglie@unict.it; 3STAR Integrated Research Unit, Università di Foggia, 71121 Foggia, Italy; matteo.francavilla@unifg.it

**Keywords:** hydrolysis, oxidation, municipal biowaste, food waste, biosurfactants, biopolymers, antifungal agents, anaerobic fermentation, compost, biotechnology

## Abstract

Municipal biowastes are conventionally treated by assessed anaerobic and aerobic fermentation to produce biogas, anaerobic digestate, and compost. Low-temperature hydrolysis and the oxidation of the digestate and compost, which are still at the experimental stage, are known to yield water-soluble value-added chemical specialities for use in different sectors of the chemical industry and in agriculture. The present paper reports the application of the two chemical reactions to the biowastes before fermentation. The products obtained in this manner are compared with those obtained from the chemical reactions applied to the fermented biowastes. Based on the experimental results, the paper discusses the expected environmental and economic benefits of the above chemical processes and products in comparison with the products obtained by other known biotechnologies for the valorisation of biomass as a feedstock for the biobased chemical industry. The results point out that a sustainable biowaste-based refinery that produces biofuel and biobased chemicals may be developed by integrating chemical and fermentation technologies.

## 1. Introduction

### 1.1. Current Research for the Valorisation of Biomass as Feedstock Producing Chemicals

The current research aims to valorise biomass alternatives to fossil feedstock for the production of fuel, chemicals, and materials [1,2,3]. Fossil feedstock exploitation is based on chemical technology. Biomass is processed by fermentation and/or biotechnology [2,4]. Figure 1A shows a simplified scheme based on biotechnology. The scheme encompasses fermentation biotechnology and thermochemical reactions. Fermentation needs biomass pretreatment [5] to separate polysaccharides from lignin, since lignin inhibits fermentation. Fermentation demolishes native polysaccharides to simple monomers. These need a second biochemical reaction to obtain biopolymers. The two fermentation steps require selected and/or genetically modified microorganisms. The lignin is burned to recover its heat value or is thermally degraded at high temperature to obtain simple molecules, which are usually referred to as building blocks [6] and are used for manufacturing fine commodities and speciality chemicals. These processes destroy the functionalities of native lignin, do not allow the exploitation of its full potential as a feedstock of value-added chemicals, cause the emission of fine dust and greenhouse gases, and/or require high energy consumption. This technology model has several gaps. The model has been applied so far to biomass, which is mostly sourced from dedicated agricultural crops. The cultivation of crops as feedstock for the production of biofuel and chemicals causes social conflict with regard to the use of agricultural areas for non-food crops [7]. The economic sustainability of the model in Figure 1A is negatively affected by the costs of the crop collection and the manufacture of the products. The process yields only building blocks and monomers, whose production cost is 2–7 times higher than the 1 EUR kg^−1^ of the commercial counterparts derived from fossil sources [1,8,9]. Additional cost is needed to process the building blocks and monomers, in order to obtain fine and speciality chemicals and finished consumer products. 

Organic chemistry offers a wide range of chemical routes for converting monomers and building blocks into a variety of molecules for the chemical industry to manufacture bulk and fine chemicals and finished consumer products that start from fossil-sourced hydrocarbons. In principle, building on current knowledge, biomass may be a potential sustainable feedstock alternative to fossil feedstock for the production of thermal and electric energy and the manufacture of basic, commodity, and fine and specialty chemicals. However, biomass requires the development of dedicated technology due to its high chemical complexity, entropy, and site-specific variability, compared to fossil sources. Together, the technological factors, biomass collection costs, social concerns regarding using crops for non-food production, and the reluctance of major chemical companies to undertake biochemical technology as a core business are hindering the development of the biobased chemical industry and preventing it from reaching the desirable commercial level [1,2,8,9,10,11].

To overcome the criticalities of the scheme in Figure 1A, the alternative technology model represented in Figure 1B has been developed [12]. Compared to Figure 1A, the model in Figure 1B is distinguished by relevant differences. First, the processes in Figure 1B have been applied to municipal biowaste (MBW) as a case study feedstock. The MBW is produced worldwide. As it is expected that two-thirds of world population will be living in urban areas by 2050, cities will be the environments most affected by the negative impact of wastes. Cities are now responsible for over 50 per cent of solid waste production and emit up to 60 per cent of greenhouse gases, contributing to pollution, climate change and biodiversity loss [13]. On the other hand, cities are a potential source of low-entropy bioorganic matter, which can be sustainably exploited as feedstock for the development of the new biobased chemical industry. In countries implementing separate-source collection practices, MBW is available in confined spaces. Unlike the case of biomass from agricultural sources, MBW is a negative-cost feedstock [14,15], as collection costs are paid off by citizens’ taxes. Therefore, MBW is a unique potential feedstock that does not present the collection cost and social concern criticalities [7] of biomass from an agricultural source. Secondly, the traditional fermentation for the scheme in Figure 1B occurs by the native microbial population and does not involve the production and maintenance of selected costly microorganisms. Thirdly, the low-temperature chemical reactions do not involve high energy consumption or the emissions of noxious secondary products. 

### 1.2. State of the Art of Figure 1B Technology

The scheme in Figure 1B encompasses two main sections, i.e., the fermentation section that produces anaerobic digestate and compost and the chemical section. The fermentation section represents the current process scheme of the Italian ACEA Pinerolese plant [16], a typical factory collecting and treating MBW by controlled fermentation to produce anaerobic digestate and compost. At the present time, the digestate is used for direct land application or is composted to produce low-value soil improvers, although it has been recognised that both the digestate and the compost may potentially be upgraded to high-value materials [3,4,11]. The chemical section produces soluble biobased chemical products (SBPs), including biosurfactants, biopolymers, and agrochemicals [12].

For the development of the model in Figure 1 [12], ACEA supplied the fermented materials, which were chemically processed by low-temperature hydrolysis and oxidation. The availability of the plant anaerobic digestate and compost allowed the addressal of the issue of the site-specific variability of biomass, which could potentially hinder the industrialisation of SBPs. The chemical composition of MBW depends on the local climate conditions and social consumption habits. This in turn may affect the chemical composition and properties of SBPs and not allow industrial manufacturers to release products with reproducible specifications and properties to the chemical market. Montoneri et al., 2022 [17] have shown that fermentation allows reducing the problem of the compositional variability of MBW collected in different seasons and locations. The authors also developed an analytical tool which was capable of predicting the chemical composition and performance of SBPs obtained from different composts. The results predict that the control of SBP quality in real industrial production is feasible. 

The development of the chemical section in Figure 1B represents the core of the past experimental work by the authors of the present paper. First, a low-temperature hydrolysis process was developed in order to obtain the SBPs, which kept the functional groups of the proximates of the pristine fermented MBW [12]. The digestate and the compost, obtained from different mixes of digestate, urban gardening residues, and sewage sludge, were hydrolysed at 80 °C and an alkaline pH. The obtained SBPs contain molecules with a molecular weight ranging from 5 to over 750 kDa. The macromolecules are constituted by aliphatic and aromatic carbon types substituted by acid and basic functional groups bonding the mineral elements of the first through to the fourth group, which are memories of the organo-mineral lignocellulose structure present in the pristine MBW. The SBPs have been proven multipurpose products for use in several sectors of the agricultural and chemical industry [11]. They have been demonstrated to be biostimulants [18] and anti-pathogen agrochemicals for the cultivation and protection of plants [12], as well as biosurfactants for the manufacture of detergents, sequestering agents for the remediation of soil polluted by trace metals and organics, and biopolymers for the fabrication of composite plastic articles [12]. The integration of an MBW treatment plant, such as the ACEA plant, with the chemical hydrolysis and oxidation processes (Figure 1B) allowed the envisioning of a municipal biowaste biorefinery that is expected to produce biofuel from the anaerobic fermentation of MBW, as well as to produce biobased SBPs by chemical hydrolysis and the oxidation of the MBW anaerobic digestate and compost. In this scenario, the SBPs were intended to be new biobased products which could potentially replace the commercial products derived from fossil feedstock.

### 1.3. Further Research and Novelty to Justify the Present Work

The aim of the work reported in the present paper was to test an alternative processing scheme for the valorisation of MBW as a feedstock, which was based on chemical technology only and therefore excluded the fermentation and/or thermochemical sections indicated in Figure 1B and/or Figure 1A, respectively. This alternative option would increase the production flexibility of the envisioned biorefinery, which is an assessed key practice in conventional fossil oil refineries [19]. For the industrialisation of the technological scheme in Figure 1B, the site-specific variability of MBW represents a critical shortcoming, as it can affect the SBPs’ chemical composition and properties and performance and not allow the marketing of products with guaranteed reproducible performance. On the other hand, the compositional variability of MBW and the derived fermentation products may allow the widening of the range of producible SBPs with different chemical compositions, properties, and performances in specific tailored applications. According to this perspective, the development and implementation of the hydrolysis and oxidation processes applied directly to the as-collected MBW, and their integration into the technological model represented in Figure 1B, would allow the envisioned biorefinery modulation of the relative production of biogas and SBPs, depending on the changes in the market demands and prices, as happens for the oil refineries [19].

According to its scope, the present paper reports the application of mild chemical hydrolysis and oxidation reactions to the as-collected MBW slurry. The experimental design comprises three different treatments: (i) the MBW centrifugation to separate the water-soluble materials from the insoluble materials; (ii) the MBW hydrolysis; and (iii) the MBW oxidation, followed by centrifugation to separate the respective water-soluble and insoluble materials. All the crude soluble materials were processed by ultrafiltration with membranes with different cutoffs in the 0.2–750 k Da ranges, in order to separate different fractions and to assess the molecular weights of the crude product composition. The separated fractions were characterised for their total C and N content, C types, and functional group composition by 13C solid state NMR spectroscopy and for their surfactants’ properties. 

The experimental plan was expected to produce SPBs with different chemical compositions and properties from the SBPs obtained from the fermented MBW in Figure 1B and therefore to widen the range of chemical specialities producible in the envisioned biorefinery. The results described hereinafter show how the SBPs obtained by the one-step chemical treatments of MBW compare with the SBPs obtained according to the multistep fermentation and chemical treatments in Figure 1B and to show what the potential impact of the envisioned biorefinery is.

## 2. Results

The as-collected MBW from the ACEA plant, which comprised unsorted food wastes, was an aqueous slurry with pH 4.8, 10% total solid dry matter, and 84.4% volatile solids (VS), with 41.7% C and 2.5% N content referring to the dry matter. The slurry was characterised for the content of the waste proximates by sequential extraction in different solvents according to the A.O.A.C. methods [20], i.e., the content of lipids soluble in cyclohexane, mono- and disaccharides soluble in plain deionised water, hemicellulose soluble and proteins soluble in 1 N aqueous HCl, cellulose soluble in 12 N aqueous HCl, and insoluble lignin. Table 1 reports the mass and the C and N content (*% w/w*) for the isolated waste proximates. 

According to the experimental design (see Section 1.3), Table 2 reports the results of the three different treatments of the MBW, i.e., the mass and C yields of the water-soluble and insoluble matter obtained by the physical centrifugation treatment only and by the chemical hydrolysis and ozonisation reactions followed by centrifugation of the hydrolysed and ozonised slurries. 

The chemical reactions of the as-collected MBW were carried out according to the experimental conditions reported in the previous work, i.e., in pH 13 water at 80 °C for the hydrolysis [12] and at room temperature for the oxidation [21] by flowing ozone through the alkaline aqueous phase (see Section 4). According to the previous work [12], MBW contains many naturally occurring metals, including transition metals. These could act as catalysts for the above chemical reactions. For example, Fe ions bonded to the SBPs obtained from MBW compost were proven to be responsible for the oxidation of many organic pollutants present in industrial wastewaters [12] and for the oxidation of ammonia to nitrogen [17]. Therefore, the hydrolysis and oxidation reactions reported in the present work were carried out in the absence of added metal catalysts.

The data in Table 2 show that the chemical reactions cause a significant increase in the recovered soluble mass and organic C, compared to the soluble mass and organic C recovered by centrifugation only of the pristine MBW. Conversely, the recovered insoluble mass and organic C obtained in the chemical reactions are significantly lower than those recovered in the centrifugation treatment. The higher amount of insoluble material recovered by centrifugation is consistent with the presence of insoluble lignocellulose material in the as-collected MBW, which is made by the recalcitrant insoluble lignin and part of the other proximates listed in Table 1, intimately linked by covalent or donor–acceptor bonds. The lignocellulose components of the as-collected MBW cannot be separated by the physical centrifugation treatment. Their intermolecular bonds are strongly affected by their concentration in water and/or prolonged chemical treatment in aqueous media at different pH levels. 

According to the previous work [12], the hydrolysis of the water-insoluble lignocellulose material present in the fermented MBW occurs at the O=C(Ar)-OR moieties (with Ar standing for aryl-alkyl and R for alkyl or polysaccharide residue), yielding hydrophilic carboxylic (Ar-COOH) and hydroxyl (ROH) functional groups. In this fashion, the reaction carried out at 60–80 °C produces high-molecular-weight soluble lignocellulose materials. At higher hydrothermal treatment temperatures, the reaction yields only low-molecular-weight small phenolic molecules [22,23,24,25,26,27]. A similar solubilisation and depolymerisation effect has been observed via the ozonisation of fermented MBW at room temperature. The oxidation occurs at the lignin aromatic chromophore moieties. The reaction causes the opening and conversion of aromatic rings to water-soluble high-molecular-weight aliphatic hydroxyl acids [21]. However, in this case, even at room temperature, over 70% of the pristine material is converted to small molecules. 

In the present work, to assess whether and how the chemical reactions affected the macro-molecularity of the native lignocellulose material of the pristine MBW, the water-soluble materials (Table 2) were further processed by sequential membrane ultrafiltration through eight polysulphone membranes with decreasing molecular cutoffs ranging from 750 kDa to 0.2 kDa. The process allowed the collection of the membrane retentates at 750 kDa (R750), 150 kDa (R150), 100 kDa (R100), 50 kDa (R50), 20 kDa (R20), 5 kDa (R5), and 0.2 kDa (R0.2) and the final permeate at 0.2 kDa (P0.2). Figure 2 reports the C recoveries relative to the total organic C of the crude soluble phases. The data are calculated from the mass yield and corresponding C content for the recovered retentates and permeate fractions given in Appendix A.

Figure 2 shows that the soluble matter obtained in the hydrolysis and ozonisation reactions contains fractions with a higher molecular weight than the fractions obtained by centrifugation only of the pristine MBW. Specifically, the R750 fractions recovered from the crude hydrolysed and ozonised soluble matter account, respectively, for 41% and 27% of the total organic C content in the respective crude products, against 5.6% C for the R750 fraction recovered from the pristine MBW treated by centrifugation only. Relevantly, the ozonisation reaction produces more low-molecular-weight organic matter than the hydrolysis reaction, as shown by the higher amount of organic C recovered with the ozonised P0.2 fraction (65%), compared to the hydrolysed P0.2 fraction (48%). This finding is consistent with the results obtained in the ozonisation [21], compared to the hydrolysis [12] of fermented MBW. 

The pristine MBW proximates (Table 1), the crude soluble and insoluble products (Table 2), and the corresponding retentate and permeate fractions (Figure 2) were analysed by solid state 13C NMR spectroscopy for the content of the C type and functional groups.

Figure 3 reports the relative composition of the C types and functional groups for each product based on the integration of the band areas in the 13C NMR spectrum measured in the falling chemical shift (δ, ppm) ranges: 0–53 for aliphatic (Af) C; 53–63 ppm for amine (NR) and methoxy (OMe) C; 63–95 ppm for alkoxy (OR) C; 95–110 ppm for anomeric (OCO) C; 110–160 ppm for total aromatic (Ph) C; and 160–185 ppm for carboxylic and amide (COX, X = OR, OM, NR, R = H, alkyl and/or aryl) C. The values given as zero mean that no signals could be distinguished from the background noise. The total integrated band area was assumed to represent the total C moles in the analysed sample (see Materials and Methods Section 4). Detailed data are also given in Appendix A. 

The 13C data show that the isolated proximates (Figure 1A) contain the main C types and functional groups characterising the lignocellulose components, i.e., OCO and OR for saccharides, NR and COX for proteins, Ph for lignin, and Af and COX (X = OR) for fats. The relative composition of the functional groups is highly consistent with the nature of the isolates. For example, all contain the OR and OCO functional groups, except the lipids. The aliphatic total aromatic C ration has the highest value for the lipids and the lowest for lignin. The protein isolate has the highest content of total NR and COX functional groups. The mono- and disaccharides have the highest content of total OR and OCO functional groups. The 13C data for the MBW proximates help to interpret the 13C data for the products and fractions in Figure 1B and C, respectively (see Section 3 below). 

## 3. Discussion

To be a competitive alternative to fossil-based refineries, the envisioned MBW-based refinery (see Section 1.3) should produce a wide range of products to meet the diversified demands of the chemical market. This requires the investigation of different feedstocks and processes. The European MBW treatment plants collect biowaste with variable composition, process it with different fermentation treatments, and produce different plant streams. A typical example is the ACEA plant, which, according to the fermentation section in Figure 1B, can produce several different composts from mixtures in different proportions of food waste digestate, gardening residues, and sewage sludge. 

A potential advantage of considering MBW as a candidate feedstock in place of fossil C sources stems from the chemical composition differences. As shown in Table 1 and Figure 3A, MBW contains all the proximates present in native biomass and is rich in functional groups containing oxygen. Fossil carbon and oil do not contain oxygen and need severe thermal and/or multistep chemical processes to manufacture the basic organic chemical commodities and a wide variety of other chemicals to manufacture the products and materials for human consumption [28,29]. A somewhat analogous approach characterise the scheme in Figure 1A, which processes biomass feedstock by destructive thermochemical and biochemical pretreatments to produce monomers that are used in the subsequent polymerisation steps. By comparison, the technological scheme in Figure 1B is designed to improve the solubility properties of the MBW lignocellulose matter and, at the same time, to maintain its macromolecular structure and functional groups as much as possible. 

For the implementation of the scheme in Figure 1B, the authors’ previous work [12] has applied the mild hydrolysis and oxidation treatments, following the fermentation pretreatments producing the anaerobic digestates and compost. In these pretreatments, the readily biodegradable lipids, saccharides, hemicellulose, and protein proximates (Table 1) are converted to biogas. Thus, at least half of structured functionalised organic matter is destructed, which otherwise could be recoverable in soluble form for the manufacture of value-added chemical specialities. In the present work, the fermentation pretreatments were skipped and the mild chemical treatments were applied directly to the as-collected pristine MBW. The scope of this strategy was to reduce the number of steps of the scheme in Figure 1B to a one-chemical step (hydrolysis or oxidation) and to assess the chemical composition of the products, in comparison to the products of the previous work, obtained from the hydrolysis [12] and oxidation [21] of the fermented MBW. In the context of the state of the art given above, the collection of the results of the present work represents a new step forward in the development and realisation of the above MBW-based refinery with the production flexibility [19] to compete with fossil feedstock refineries.

### 3.1. The Chemical Composition of Pristine, Hydrolysed, and Ozonised MBW

The 13C data in Figure 3 are mostly valuable for assessing the chemical composition of the products obtained in the present work. The data (Figure 3A) for the MBW proximates (listed in Table 1) are useful for interpreting the spectra of the crude products (listed in Table 2) and the R750 fractions (listed in Figure 2). The spectrum of each proximate (Figure 3A) is characterised by the major signal/s confirming the presence of the most abundant C type/s expected for the proximate, i.e., the 92.7% Af and 3.7% COX C for the lipids; 38.7% OR, 28% Af, 12.1 COX, 8.8% NR, and 4.1% OCO C for the sugars, amino sugars, and acetyl amino sugars; and 34.9% Af, 22.4% COX, 16.1 OR, 6.39 NR, 4.3% OCO C for the hemicellulose and proteins. Lignin is characterised by the highest content of total aromatic (Ph) C, at 22%, compared to the other proximates. However, lignin also contains high amounts (36%) of OR C, which is likely contributed by the aryl alkyl ether linkages, where the alkyl portion may be an aliphatic C chain or alkoxy functional group in a cellulosic moiety. In essence, all the isolated proximates have in common many of the functional groups listed in Figure 3 and differ from one another according to the relative abundance of these groups. This situation is typical of the complex composition of the native lignocellulose material present in biomass. 

Similar features are exhibited by the 13C data for the crude products (Figure 3B). The most abundant C in the “centrifugation crude insoluble matter” obtained from the pristine MBW is the OR C (48.1%). This is coupled to the OCO C (9.8%) and suggests the presence of water-insoluble cellulose as a major component. The 5 OR/OCO ratio for this product corresponds to a glucose unit. By comparison, for the “centrifugation crude soluble matter” obtained from the pristine MBW, the most abundant C are the Af (32.6%), OR (28.9%), and COX (17.1%) C. These features coupled to the 2.92% OCO C % and 6% NR suggest the presence of water-soluble hemicellulose and protein matter as major components. The separated water-insoluble and water-soluble materials contain, respectively, 5% and 8% aromatic C. The presence of aromatic C in both the water-insoluble and water-soluble matter suggests the presence of different organic moieties, where lignin is bonded to the respective cellulose and hemicellulose-protein major components of the pristine MBW insoluble and soluble matter.

The 13C data (Figure 3B) for the hydrolysis and ozonisation of the insoluble and soluble matter exhibit very different chemical compositions, from each other and from the pristine centrifugation of the insoluble and soluble matter. The C distribution pattern of the “hydrolysis crude insoluble matter” indicates a more complex lignocellulose matter, compared to the patterns for the pristine MBW proximates (Figure 3A) and for the MBW “centrifugation crude insoluble and soluble matter” (Figure 3B). The pattern for the “hydrolysis crude soluble matter” is much simpler. It is composed by 52% Af, 16% COX, 12% OR, and 10% total aromatic C. Compared to the “centrifugation crude insoluble and soluble matter” and the “hydrolysis crude insoluble matter”, the “hydrolysis crude soluble matter” is distinguished mostly by its having the highest content of aliphatic C. This feature coupled to the presence of the COX and OR functional groups suggests that the major component of the “hydrolysis crude soluble matter” is an aliphatic alkoxy/hydroxyl-carboxyl moiety. The 10% aromatic C corresponds to one lignin-like aromatic ring every 31 aliphatic C moles. 

Compared to the centrifugation and hydrolysis crude products, the 13C spectra of the ozonisation crude products are characterised by the three most relevant features. The “ozonisation crude soluble matter” (Figure 3B) contains much more carboxyl (COX) C (22.7%). The “ozonisation crude insoluble matter” contains the same amounts of OR and OCO C as the “centrifugation crude insoluble matter”. Both the “ozonisation crude insoluble and soluble matter” do not contain detectable aromatic C. Apparently, complete or extensive oxidation of the aromatic C present in the pristine MBW lignocellulose material is achieved by the ozonisation reaction. The data are consistent with the production of a highly carboxylated soluble material identified by the 37% Af, 23% COX, 9.6 OR, and 3 OCO C measured for the “ozonisation crude soluble matter” and a residual insoluble cellulose-like and/or polysaccharide matter identified by 46.3% OR and 9.2% OCO C and 24% Af, 8% NR, and 8% COX measured for the “ozonisation crude insoluble matter”. 

### 3.2. Molecular Weight Distribution and Chemical Composition for Centrifugation, Hydrolysis, and Ozonisation Crude Soluble Products

For the scope of the present work, the crude soluble products obtained by the physical centrifugation and in the two chemical treatments of the as-collected pristine MBW needed further investigation. They contained lignocellulose matter, which by virtue of its solubility property in water could be more valuable than that contained in the insoluble products for the manufacture of value-added chemical specialities. The molecular weight distribution in the crude soluble products was a key issue of the present investigation. The data in Table 2 and Figure 2 show that the crude soluble matter constituting 26% of the total pristine as-collected MBW mass separated by centrifugation (Table 2) contained mostly low-molecular-weight compounds, as shown by the 88.7% P0.2 C content (Figure 2). Conversely, the crude soluble matter constituting, respectively, 54% of the total hydrolysed MBW mass and 47% of the total ozonised MBW mass (Table 2) contained much more polymeric material, i.e., 42% and 28% of the total organic C recovered with the R750 and R100 of the “hydrolysis and ozonisation crude soluble product”, against 5.6% C for the total organic C recovered with the R750 fraction of the “centrifugation crude soluble product”. 

As they are largely the principal high-molecular-weight fractions, the R750 and R100 fractions were analysed by 13C NMR spectroscopy. The 13C data for these fractions (Figure 3C) reflect the C distribution pattern for their pristine crude soluble matter (Figure 3B), although with some relative quantitative differences. Particularly interesting is the fact that both the “ozonisation soluble R750” fraction (Figure 3C) and the “ozonisation crude soluble matter” (Figure 3B) contain no aromatic C, whereas the corresponding hydrolysis and centrifugation soluble R750 fractions (Figure 3C) and crude soluble matter (Figure 3B) do contain aromatic C. This is a further confirmation that the ozonisation reaction can achieve the complete oxidation of the aromatic C present in the pristine MBW lignocellulose material and yield a high-molecular-weight aliphatic carboxylate water-soluble material. It seems that this material is present in a more concentrated form in the “ozonisation soluble R750” fraction (Figure 3C). Indeed, compared to the “ozonisation crude soluble matter” (Figure 3B), the “ozonisation soluble R750” fraction (Figure 3C) contains 92% total Af, COX, and OR C (Figure 3C) versus 69% for the “ozonisation crude soluble matter” (Figure 3B). This allows a better tentative identification of the high-molecular-weight water-soluble aliphatic carboxylate material produced by the ozonisation reaction. The 13C data for the “ozonisation soluble R750” fraction (Figure 3C) correspond to a polymeric alkoxy/hydroxy-carboxyl compound with a molecular weight ≥750 kDa and the virtual repeating unit -(CH_2_)_4.1_-[CH(OR)]_0.5_-COX-. Similarly, the 13C data for the “ozonisation soluble R100” fraction (Figure 3C) are consistent with the high concentration of a polymer with a molecular weight in the 100–750 kDa range and a virtual repeating unit -(CH_2_)_1.3_-CH(OR)-[CH(NH-)]_0.5_-COX-.

The production of poly-hydroxyalkanoate from MBW has been reported following the use of a mixed microbial culture [30]. The polymer was found to have a molecular weight of 850 kDa and a 53/47 *w*/*w* ratio of the hydroxybutyrate/hydroxyvalerate repeating unit. The product was obtained in two stages. In the first stage, the acidic fermentation of MBW produced the substrate, which was fermented in the second stage in the presence of selected poly-hydroxyalkanoate-producing bacteria. The product was characterised by a 13C solid state NMR spectroscopy and molecular weight by HP-SEC/TDA measurement, using a commercial sample of poly-hydroxyalkanoate as an analytical reference. The reported 13C spectra and molecular weight of the reference material and the product obtained in the two-stage fermentation treatment of MBW are highly consistent with the 13C data and molecular weights for the “ozonisation soluble R750” and the “ozonisation soluble R100” fractions obtained in the present work. To the authors’ knowledge, the high-molecular-weight hydroxyalkanoate-like polymers obtained by the one-step chemical oxidation of MBW, as performed in the present work, have never been reported. 

### 3.3. Depolymerisation of the MBW Proximates in the Chemical Reactions

Whereas the high-molecular-weight hydroxyalkanoate-like polymers obtained by ozonisation represent an innovation with potential added-value for the envisioned MBW-based biorefinery, the formation of the large amount of P0.2 low-molecular-weight co-product (Figure 2) indicates that depolymerisation constitutes a shortcoming of the proposed chemical reactions. For the scope of the present work, the formation of low-molecular-weight products constitutes a loss of potentially valuable organic matter. The same drawback was observed in the chemical reactions applied to the MBW anaerobic digestate and compost according to the scheme in Figure 1B, where ozonisation [21] caused higher depolymerisation than hydrolysis [12]. The two reactions may involve two different plausible mechanisms. Hydrolysis occurs at the lignocellulose O=C(Ar)-OR moieties yielding carboxylic (Ar-COOH) and hydroxyl (ROH) functional groups [12]. Ozonisation occurs at the aromatic rings and causes ring opening and the conversion of aromatic C to aliphatic hydroxyl-carboxyl C [21]. 

Several researchers have studied the ozonisation of different compounds and materials, including proximates that are similar to those dealt with in the present work (Table 1). Oxidation by ozonisation has been reported to occur primarily at unsaturated carbon, such as in lignin aromatic moieties, and unsaturated aliphatic carbon chains, and to produce aliphatic carboxylic acids [31,32,33,34]. Polyamide bonds and saturated C of protein main chains are not degraded [35]. Saturated fatty acids do not react [36]. Ozonisation is practised in the paper and pulp industry, to delignify and bleach cellulose [37]. Ozonisation of plant biomass has shown that wood destruction by ozone is accompanied by the degradation of lignin, hemicelluloses, and cellulose [38]. In this case, it has been found that upon increasing the ozone consumption, the content of residual lignin in the wood sample decreased from 25.3% to 1.7% and the content of cellulose (41–46%) remained at the level in the original wood, although the cellulose polymerisation degree decreased from 700 to 200. Similar depolymerisation has been reported for the ozonisation of polysaccharides [39] with various reaction times (0, 15, 30, 45, and 60 min). In this case, the main chain of the pristine polysaccharides was preserved, but the molecular weight of the ozonised samples decreased with the increase in reaction time. The results were ascribed to the generation of hydroxyl radicals in water and the lysis of the b-d-(1,4)-glucosidic linkages of the polysaccharides. The products’ depolymerisation degree depended on the rate or amount of ozone passed through the reaction mixture and its pH [40].

Consistently with the results published in the literature cited above, the data collected in the present work confirm that oxidation by ozonisation occurs primarily and extensively at the lignin aromatic rings and produces soluble aliphatic hydroxyl-carboxylic acids (Figure 3B,C) containing no aromatic C, while the “ozonisation crude insoluble matter” (Figure 2B) contains mainly polysaccharide cellulose-like matter. Based on the published results given above, the higher depolymerisation by ozonisation than by hydrolysis, which has been observed in the present work, is likely due to the fact that oxidation by ozonisation occurs at both the aromatic C=C [39] and the C-O-R [39,40] bonds of the organic O=C(Ar)-OR moieties constituting the polymeric lignocellulose matter of the as-collected pristine MBW, while hydrolysis causes only the lysis of the C-O-R bonds. 

### 3.4. One-Step Chemical Treatment vs. Multistep Fermentation/Chemical Treatments of MBW

#### 3.4.1. Comparing Products’ Composition and Yields

Figure 4 summarises the results of the 13C NMR spectra recorded in the previously cited work [12,21] for the SBPs obtained according to the multistep fermentation–chemical treatments depicted in Figure 1B. The measurements for the data reported in Figure 4 were performed as described for Figure 3 (Section 2) and in Section 4. 

The data refer to the hydrolysed and ozonised products obtained from the pristine MBW fermented materials, i.e., the anaerobic digestate and the composts obtained from urban green residues (CG) and from a mix of green residues and anaerobic digestate (CGV). The sequential membrane ultrafiltration of the hydrolysis and ozonisation crude soluble product obtained from the anaerobic digestate was performed as described for Figure 2. For the hydrolysis and ozonisation crude soluble products obtained from the composts, the sequential membrane ultrafiltration was performed starting with the 35 kDa cutoff membrane. More details are given in Appendix A. 

To compare the products obtained in the previous [12,21] work by the multistep fermentation–chemical treatment (Figure 1B) with the products described in Section 3.1, Section 3.2 and Section 3.3, it should be acknowledged that the MBW digestate and composts are quite different from the pristine as-collected MBW. In the anaerobic fermentation step depicted in Figure 1B, the most readily degradable proximates (lipids and saccharides in Table 1) are converted to biogas. In the aerobic fermentation, more degradable biomass is mineralised, leaving a recalcitrant lignin-like residue that constitutes the compost. Figure 4A shows that the “hydrolysis soluble R35 fractions from the two CG and CGD composts” have indeed less aliphatic and more aromatic C than the “hydrolysis soluble R750, R150 and R100 fractions from the anaerobic digestate”. By comparison, Figure 4B shows that ozonisation reduces the differences between the fractions obtained from the composts and the anaerobic digestate, with the major difference being the drastic reduction in the aromatic C in the fractions obtained from the compost. These results are highly consistent with the different mechanisms of the two chemical reactions described in Section 3.3. It is quite evident that, while hydrolysis of the fermented MBW causes only the lysis of C-O-R bonds in the O=C(Ar)-OR moiety representing the virtual repeating unit of the polymeric lignocellulose matter of the digestate and composts, and therefore yields soluble products maintaining the different aromatic C content of the pristine digestate and composts, the ozonisation occurring at the aromatic C=C bonds is capable of converting the recalcitrant lignin-like matter present in the pristine composts to aliphatic C chains. 

Measurement by size exclusion chromatography–multiangle light scattering analysis showed that the “ozonisation soluble R35 from the CGD and CG composts” had average molecular weights of 487 and 223 kDa, respectively, which were higher than the measured 279 and 18 kDa values for the corresponding “hydrolysis soluble R35” from the two composts. However, while the “hydrolysis soluble R35” fractions accounted for over 90% of the total C recovered by the sequential membrane ultrafiltration, the “ozonisation soluble R35” fractions accounted for only 6–25% of the total recovered C, with the residual C accounted for by fractions with molecular weights ranging from 20 kDa to <5 kDa. The data [21] for the “hydrolysis and ozonisation soluble” fractions obtained from the “anaerobic digestate” confirmed the same compositional and yield trends of the products as those observed for the composts. 

These results are consistent with those obtained in the present work for the one-step chemical treatment of the as-collected pristine MBW by hydrolysis and ozonisation. In essence, the oxidation by ozonisation of the lignocellulose organic matter in the as-collected pristine MBW and in the fermented MBW materials yields higher depolymerisation than the hydrolysis reaction, although the ozonisation high-molecular-weight fractions contain a higher concentration of alkoxy/hydroxy-carboxyl biopolymers. It appears, therefore, that ozonisation can yield quite similar polyalkonoleate-like polymers, in spite of the chemical composition differences of the pristine lignocellulosic matter from the different sources. 

#### 3.4.2. Comparing Product Performance

The previously cited work [12,21] pointed out the high potential of the R750, R150, and R100 fractions as new value-added chemical specialities, on the basis of their macromolecular structure and the surfactants’ properties. Surface tension measurements allow the screening of products for their potential as surfactants and active principles to manufacture finished product formulates for the consumers’ general use. Figure 5 and Appendix A report the surface tension activity measured in water containing 2 g L^−1^ of product for the high-molecular-weight fractions obtained in the present work (Figure 3C) by the one-step chemical treatment and in the previous work (11, 19) by the multistep fermentation–chemical treatment of the as-collected MBW (Figure 1B).

The data indicate that all the products have the capacity to lower the plain water surface tension from 70 to 60–37 mN m^−1^. The products may be roughly distinguished in two further sub-classes, characterised by different γ values, i.e., the first with 60–50 γ and the second with γ < 50. The “centrifugation soluble R750” and the “hydrolysis soluble R750” (Figure 3C) obtained by the one-step treatment belong to the second sub-class, together with the “ozonisation soluble R750, R150 and R100” (Figure 4B)” obtained from the anaerobic digestate in the multistep fermentation–chemical treatment. The same classification based on surface tension values measured at a 2 g L^−1^ added product concentration in the water solution was found to apply to the high-molecular-weight fractions isolated from the hydrogenated hydrolysates obtained from the anaerobic digestate and the CGD compost [41]. This reaction was also carried out in the mildest possible experimental conditions, i.e., in water at pH 10 and 100 °C, with no added metal catalysts and with flowing H_2_ through the aqueous solution for 2–30 min. Only the R750 fraction of the hydrogenated crude soluble hydrolysate obtained from the anaerobic digestate exhibited a γ of about 50 mN m^−1^. All the other R750 and/or R150 and R100 fractions of the hydrogenated crude soluble hydrolysates obtained from the anaerobic digestate and/or the CGD compost yielded a γ > 50. 

A complete assessment of the surfactant properties of each product requires the measurement of the surface tension at different concentrations of the added product in solution and the examination of the plot of the γ vs. the product concentration in the examined solution. These measurements allow the calculation of the critical micellar concentration (cmc), a second important parameter with which to estimate the potential surfactant’s performance in real operational conditions. At the cmc, the investigated surfactant reaches the minimum γ value, beyond which the trend of the solution surface tension vs. the added surfactant concentration is more or less flat, compared to the steeper slope of the plot at concentrations lower than the cmc. For the products obtained in the previous work, the cmc values were found to fall in the 1–3 g L^−1^ concentration [12,21]. The γ measurement of the products obtained in the present work was carried out at a 2 g L^−1^ concentration, in order to be compared with the γ values recorded for the products obtained by the multistep fermentation–chemical treatments. 

A sound comparative evaluation of the potential of the products in Figure 5 to perform in real operational conditions must also take into consideration that the surface tension and cmc values depend strongly on the chemical composition of the product and that, in turn, the chemical composition depends strongly on the parameters of the experimental conditions under which the products are obtained. For example, the 36.8 γ value shown for “ozonisation soluble R150 from anaerobic digestate” (Figure 5) has been measured for the product obtained after flowing ozone for 64 h. At the longer 88 h’ ozonisation time, the recorded γ value for the isolated R150 was 58.6. The replication of the 64 h’ ozonisation gave the “ozonisation soluble R150 from anaerobic digestate” with a measured 45 γ value. These findings point out that the ranking order of the products shown in Figure 5 based on surface tension values may undergo drastic changes, depending on the experimental conditions under which each of the two chemical reactions is carried out. The above variability of the surface tension measurements may point out that the products obtained under the same experimental conditions may not exhibit a reproducible performance in real operational conditions. This brings about the important issue of manufacturing a product with constant reliable specifications and performance in the context of scaling up the process and products to an industrial and commercial level. A similar issue has been discussed for SBPs obtained by the hydrolysis of composts with different compositions [17]. In this case, the SBPs were used as auxiliaries for reducing the ammonia production in the anaerobic fermentation of different food wastes with significant compositional differences. The results confirmed that, although the SBP composition was affected by the different compositions of the pristine compost from which they were obtained, their performance in a real operational environment was not critically compromised. 

#### 3.4.3. Comparing Products Potential for Improved Performance and Marketability

The surface activities (Figure 5) of the products obtained by the one-step and the multistep treatments are very important with regard to their entrance into the chemical market. Regarding their surfactant properties, the soluble products obtained in the multistep treatments have been proven to perform as eco-friendly potential active principles for the formulation of detergents, textile dyeing baths, dispersants, flocculants, and binding agents for the manufacture of ceramics, emulsifiers, and auxiliaries for soil/water remediation following pollution by metals and organics and for enhanced oil recovery and nanostructured materials for chemical and biochemical catalysis [12]. On the other hand, by virtue of their high molecular weight, these products have been investigated for their potential to fabricate new biobased plastic articles [12].

In the plastics field, the soluble crude products, obtained from the hydrolysis of anaerobic digestate, and from the hydrolysis of CG and CGD composts, have been used as fillers in the fabrication of composite films together with commercial synthetic polymers obtained from fossil sources [12]. However, the soluble hydrolysates exhibited several drawbacks. The most important one is the absence of melting and film-forming properties. For these reasons, they were compounded with other commercial synthetic polymers, such as poly (ethylene-co-vinyl alcohol) [12]. In this fashion, composite plastic films could be obtained by melt extrusion. Compared to the neat synthetic polymers, the mechanical properties of the composite films were found to deteriorate upon the increasing of the content of the digestate or compost hydrolysates. No useable films could be obtained from blends containing more than 10% filler. The reason for the poor mechanical property of the blends was ascribed to the content of lignin-like aromatic moieties in the filler. These were the main contributors of the increasing mechanical rigidity and decreasing mechanical flexibility upon the increasing of the filler content in the composite material. On the other hand, the biosurfactant properties exhibited by the products in Figure 5 offered possible ways to improve the manufacturing process and properties of the composite blends with a higher filler content. By virtue of the presence of lipophilic and hydrophilic C moieties, the above SBPs can interact in different ways with the synthetic polymer in the blend. These interactions may improve the compatibility of the interface between the synthetic polymer and the biopolymer filler, yield more homogeneous composites, and improve the blends’ extrudability. In this context, the SBPs obtained by the one-step chemical treatments have the same potential as the SBPs obtained by the multistep fermentation–chemical treatments. Due to the absence of aromatic C, the soluble R750 and R100 fractions (Figure 3C), isolated from the “ozonisation crude soluble matter” obtained in the one-step ozonisation treatment applied to the as-collected MBW, show that these products may have the highest potential for the manufacture of plastic articles, compared to all others obtained in the present and previous work [12,21].

#### 3.4.4. Potential Impacts from the Industrialisation of the One-Step and Multistep MBW Treatments

The optimisation and scale-up of the results reported in the present work have a high potential environmental, economic, and social impact. At least 2000 MBW treatment plants in Europe can benefit from one-step and/or multistep MBW treatment technology. These plants process a total of 35 Mt yr^−1^ MBW by anaerobic and aerobic fermentations, yielding biogas, anaerobic digestate, and/or compost [12,42]. They are not cost-effective [12]. MBW processing costs by the current conventional fermentation treatments are covered mostly by citizens’ taxes. The other two-thirds of the total 100 Mt yr^−1^ of EU-produced MBW is landfilled. They generate GHG emissions from uncontrolled waste fermentation, with an estimated 12,000 CH_4_ Mm^3^ yr^−1^ in total emissions. Applying the one-step and/or multistep MBW treatments (Figure 1B) to all EU MBW may yield up to 56 Mt yr^−1^ of new value-added SBPs chemical specialities (e.g., biosurfactants and biopolymers), with a potential 1500–150,000 EUR t^−1^ sale value [12,43]. The realisation of these economic perspectives can incentivise the conversion of the current MBW treatment plants into cost-effective biorefineries. The social impacts are the potential relief or more efficient use of the current citizens’ taxes paid for MBW disposal, a healthier living environment due to a reduction in GHGs from landfill sites, and the creation of new jobs. The new waste-based biorefineries are expected to generate a business worth 1–3 orders of magnitude more than its cost, to encourage the increase in MBW processing plants from 2000 to 5000–6000, to favour the dismissal of landfill sites, and to promote the creation of new jobs [44] along the value chain encompassing the MBW feedstock and the SBPs. The social acceptance of one-step technology would be boosted by the foreseen GHG emissions reduction [9,12], the dismissal of MBW landfill sites, and the replacement of synthetic chemical products from fossil sources with the value-added SBPs produced by the MBW-based biorefineries. The above impacts may be fully achieved by the different actors’ joint ventures (JVs), including MBW treatment companies and chemical companies producing and marketing the finished products containing the SBPs as active principles. JVs may overcome the criticalities in the feedstock supply, process engineering, product regulation, manufacturing scale, and marketing of the biobased products. 

## 4. Materials and Methods

### 4.1. Materials

The as-collected municipal biowastes (MBWs) were supplied by Acea Pinerolese SpA as a 10% dispersion in water. This material was treated by three procedures. In the first, the as-collected MBW was treated with a laboratory centrifuge operated at 5000 rpm to separate the water-soluble mass from the insoluble mass (Table 2). In the second, the as-collected MBW was added along with KOH at pH 13 and heated at 80 °C. In the third, the as-collected MBW was treated by flowing ozone through the slurry. The hydrolysed and ozonised slurries were centrifuged as described above to separate the corresponding soluble and insoluble masses (Table 2).

### 4.2. Details of the Chemical Reactions

#### 4.2.1. Hydrolysis

The as-collected MBW slurry (3066 g) and 184 g of 90% KOH at pH 13 under stirring were heated to 80 °C in 45 min and kept at this temperature for 20 min. At the end of the reaction, the separated soluble and insoluble masses were weighed. 

#### 4.2.2. Ozonisation

The as-collected MBW slurry (500 g) was diluted with 1000 mL water. KOH was added to the slurry until pH 10 was reached. Ozone was then flowed through the slurry according to previous work [21], i.e., 60 L h^−1^ oxygen with 4 mole/mole % ozone content was flowed for 38 h. During this time, 24 g KOH was added to the slurry to keep the starting pH 10 constant. At the end of the reaction, the separated soluble and insoluble were weighed. 

#### 4.2.3. Products’ Isolation and Characterisation

The separated soluble masses obtained by the three centrifugation, hydrolysis, and ozonisation treatments were further processed by sequential membrane ultrafiltration through 8 polysulphone membranes with decreasing molecular cut to collect the retentates at 750 kDa (R750), 150 kDa (R150), 100 kDa (R100), 50 kDa (R50), 20 kDa (R20), 5 kDa (R5), and 0.2 kDa (R0.2) and the final permeate at 0.2 kDa (P0.2). The obtained retentate and permeate fractions were dried at 60 °C to a constant weight and analysed for their volatile solids, ash, C, and N contents. The products were characterised for their relative C type and functional composition by 13C solid state NMR spectroscopy and for their surface tension in a water solution at a 2 g L^−1^ added product concentration. 

### 4.3. Analytical Methods

#### 4.3.1. Proximates Analysis

The as-collected MBW was treated by sequential extraction with different solvents according to the A.O.A.C. methodology [20], i.e., cyclohexane to separate the lipids soluble in cyclohexane, the mono- and disaccharides soluble in plain deionised water, the hemicellulose and proteins soluble in 1 N aqueous HCl, the cellulose soluble in 12 N aqueous HCl, and insoluble lignin. The extracts were dried and the obtained products were weighed and analysed for the content by 13C NMR spectroscopy. 

#### 4.3.2. 13C Solid State NMR Spectra

The spectra were recorded as previously reported [11,19], at 67.9 MHz on a JEOL GSE 270 spectrometer (JEOL Ltd., Tokyo, Japan) equipped with a Doty probe. The cross-polarisation magic angle spinning (CPMAS) technique was employed, and for each spectrum, about 104 free induction decays were accumulated. The pulse repetition rate was set at 0.5 s and the contact time at 1 ms; the sweep width was 35 KHz, and MAS was performed at 5 kHz. Under these conditions, the NMR technique provides quantitative integration values in the different spectral regions. Thus, the relative composition of C types and functional groups for each product in Figure 3 is based on the integration of the band areas in the 13C NMR spectrum falling in the chemical shift (δ ppm) ranges: 0–53 for aliphatic (Af) C; 53–63 ppm for amine (NR) and methoxy (OMe) C; 63–95 ppm for alkoxy (OR) C; 95–110 ppm for anomeric (OCO) C; 110–160 ppm for total aromatic (Ph) C; and 160–185 ppm for carboxylic and amide (COX, X = OR, OM, NR, R = H, alkyl and/or aryl) C. The total integrated band area was assumed to represent the total C moles in the analysed sample. All the other analytical and product characterisation details were as previously reported [12,21]. 

## 5. Conclusions

Significant innovation and advantages seem to be possibly attained by integrating the one-step chemical technology proposed in the present work in the multistep fermentation–chemical model depicted in Figure 1B. Table 3 summarises the gaps and advantages of the three technology models under consideration for the valorisation of biomass as a feedstock for the production of biofuel and biobased chemicals. 

The models encompassing the one-step chemical and the multistep traditional fermentation–chemical technology (Figure 1B) may overcome all the shortcomings of the fermentation biotechnology–incineration–pyrolysis model depicted in Figure 1A. The major gap of the one-step chemical and Figure 1B models lies in the site-specific variability of MBW. The negative effect of the poor reproducibility of SBPs’ quality and performance has been proven not significant in the use of SBPs as additives in the anaerobic fermentation of biowastes to reduce ammonia emissions [17]. For the product uses listed in Figure 1B, the effect of the MBW variability has not yet been investigated systemically. Compared to the Figure 1B multistep model, the one-step chemical model offers several advantages. The polyhydroxyalkenoleates obtained by the direct ozonisation of the as-collected MBW (Figure 3B), by virtue of their low content of aromatic C, are expected to yield composite plastic articles with higher workability and mechanical properties (see Section 3.4.3), compared to the products obtained according to the multistep Figure 1B model. Moreover, the one-step chemical model would yield higher SBPs’ productivity and allow higher production flexibility for the envisioned biorefinery integration of both the one-step chemical and the multistep model depicted in Figure 1B. These perspectives, and the SBPs potential market values and expected environmental and socio-economic benefits offer a strong incentive for further work to optimise the production, properties, and performance of SBPs in real industrial operation conditions. 

## Figures and Tables

**Figure 1 molecules-28-07670-f001:**
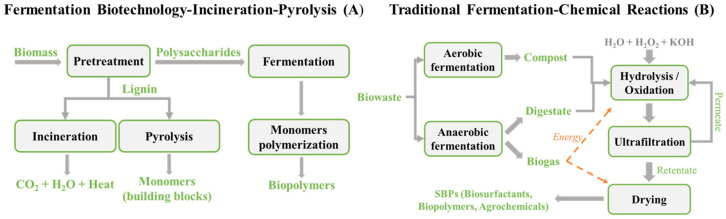
Fermentation biotechnology–incineration–pyrolysis (**A**) vs. traditional fermentation integrated with mild chemical reactions (**B**).

**Figure 2 molecules-28-07670-f002:**
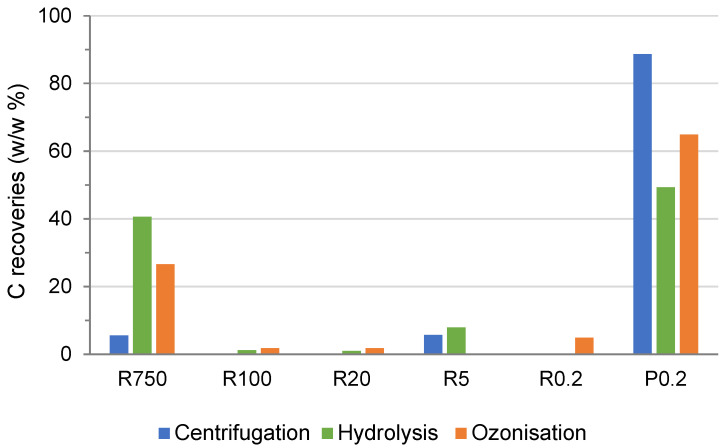
C recovery yields (% *w*/*w* referred to dry matter) for fractions isolated by membrane ultrafiltration of crude soluble matter obtained by the centrifugation, hydrolysis, and ozonisation treatments listed in Table 2, i.e., the retentates at 750 kDa (R750), 100 kDa (R100), 20 kDa (R20), and 0.2 kDa (R0.2) and the final permeate at 0.2 kDa (P0.2).

**Figure 3 molecules-28-07670-f003:**
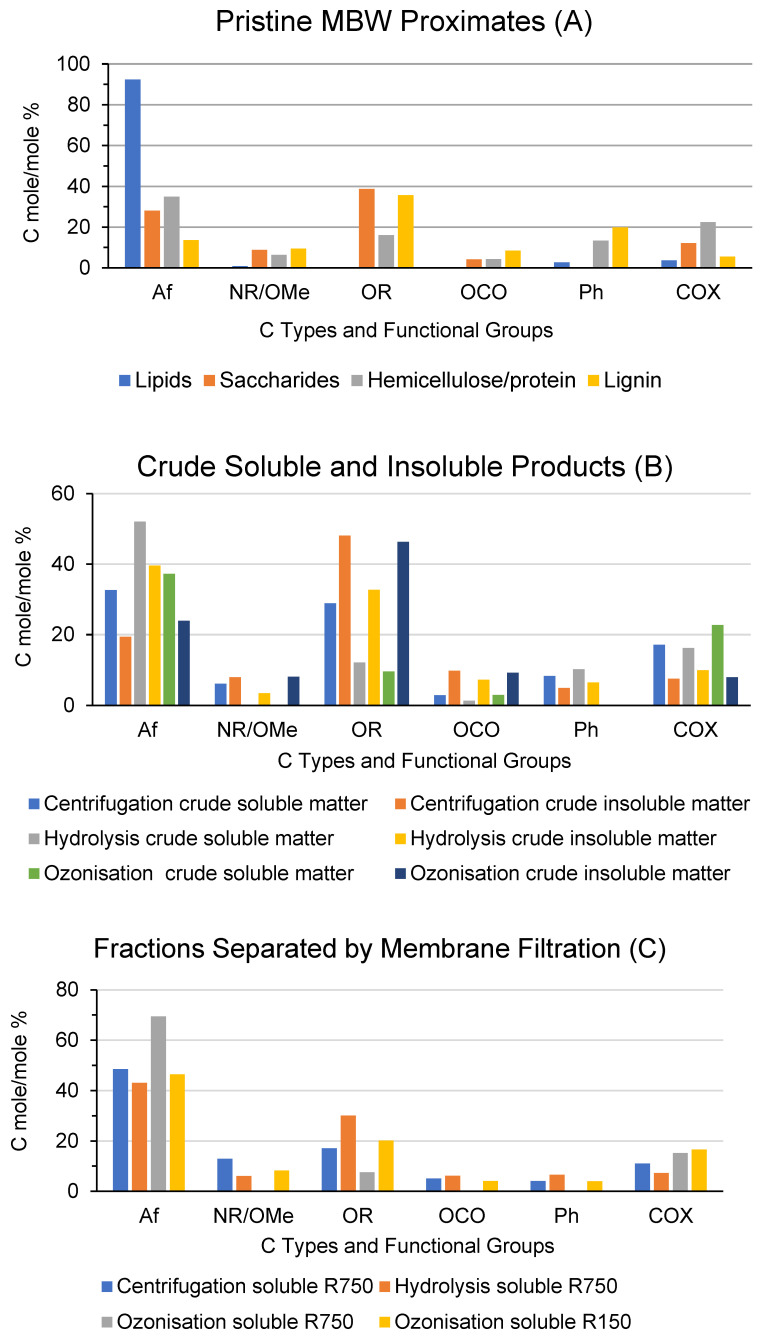
C types and functional groups composition as % mole/mole, relative to total organic C for materials obtained in the present work: (**A**) for the MBW proximates listed in Table 1, (**B**) for the crude soluble and insoluble products listed in Table 2, and (**C**) for the retentate and permeate fractions of the crude soluble products listed in Figure 2.

**Figure 4 molecules-28-07670-f004:**
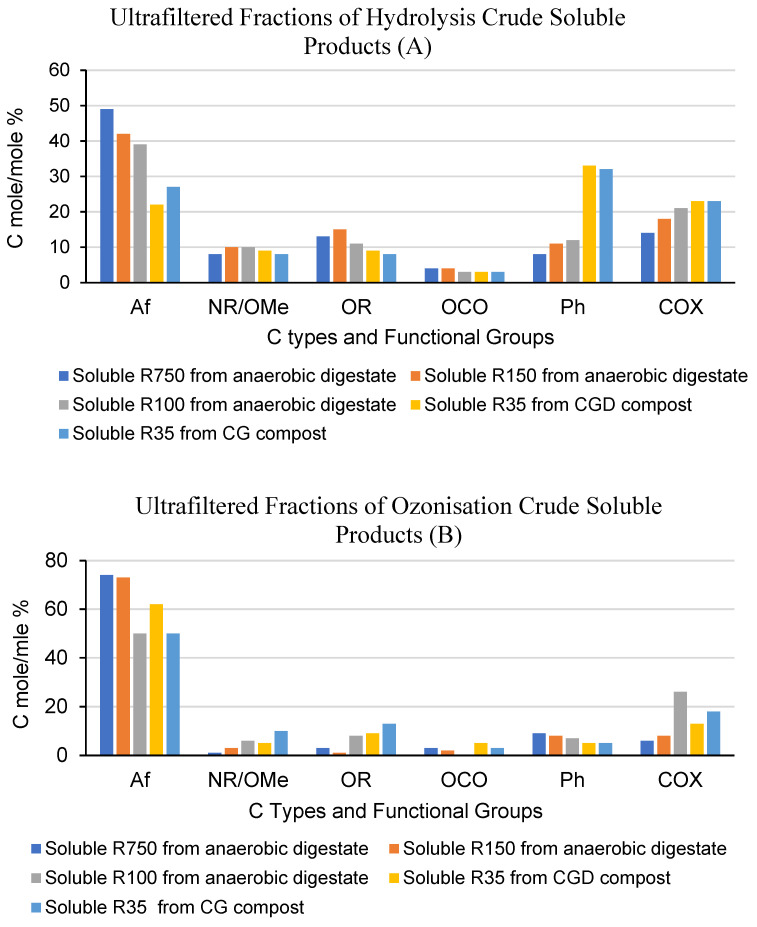
C type and functional group relative compositions as % mole/mole relative to total organic C for the retentate and permeate fractions isolated by ultrafiltration of the crude soluble products obtained by hydrolysis (**A**) of the MBW anaerobic digestate and composts (according to the scheme in Figure 1B), and by ozonisation (**B**) of the of the corresponding crude soluble hydrolysates.

**Figure 5 molecules-28-07670-f005:**
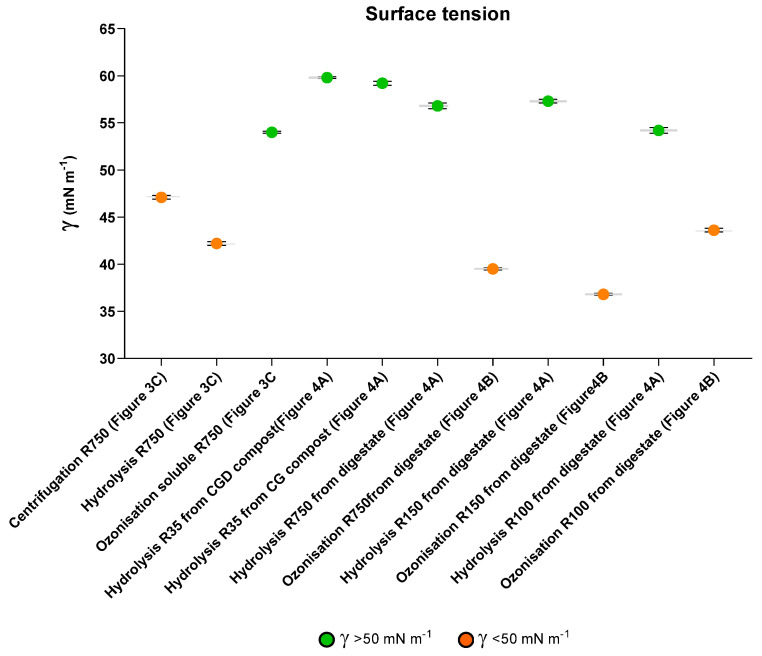
Surface tension (γ, mN m^−1^) of products in Figure 3 and Figure 4.

**Table 1 molecules-28-07670-t001:** Mass, C and N content (% *w*/*w*) ^a^ of MBW proximates.

Proximates	Mass (*% w/w*)	C, %	N, %	C/N
Lipids	13.0 ± 1.8 ab	77.1 ± 0.9 a	0.18 ± 0.04 a	425
Mono- and disaccharides	45.6 ± 5.4 c	36.6 ± 1.8 b	2.97 ± 0.13 b	12.4
Hemicellulose and proteins	18.1 ± 6.1 b	30.0 ± 9.9 b	3.89 ± 0.49 cb	7.7
Cellulose	6.2 ± 0.2 a	53.8 ± 3.1 c	3.25 ± 1.07 b	16.6
Lignin	17.2 ± 4.2 ab	42.4 ± 3.2 b	1.75 ± 0.42 b	24.2

^a^ In each column, values followed by different letters are significantly different at *p* < 0.01 level.

**Table 2 molecules-28-07670-t002:** Mass and C balance for the centrifugation treatment and for the hydrolysis and ozonisation reactions of the as-collected MBW. Data are quoted as *% w/w* ^a^ and refer to dry matter.

Materials Yield (*% w/w*)	Centrifugation	Hydrolysis	Ozonisation
Soluble mass	25.9 ± 1.8 a	54.5 ± 3.1 b	46.6 ± 1.6 c
Insoluble mass	74.1 ± 1.8 a	45.5 ± 1.2 b	45.5 ± 2.5 b
Soluble C	27.1 ± 0.4 a	40.3 ± 0.2 b	37.1 ± 2.4 b
Insoluble C	74.4 ± 0.9 a	37 ± 1.9 b	62.2 ± 3.9 c

^a^ In each row, values followed by different letters (a–c) are significantly different at *p* < 0.01 level.

**Table 3 molecules-28-07670-t003:** Gaps and advantages of the one-step chemical technology compared to Figure 1A,B multistep models.

Gaps Figure 1A Model	Gaps Figure 1B Model	Gaps One-Step Chemical Model
Biomass collection costs	none	none
Destruction of biomass chemical structure	partial	none
Fine dust and greenhouse gas emissions	none	none
High energy consumption	none	none
Social conflict	none	none
High product cost	low	low
Gaps for Figure 1B and one-step chemical models	Figure 1B model	One-step chemical model
Site-specific variability of MBW chemical composition	Potential risk of poor reproducibility of SBP quality and performance; risk mitigation is feasible, but must be assessed for each specific application, [17] and Section 3.4.2	
Advantages for Figure 1B and one-step chemical models	Figure 1B model	One-step chemical model
Site-specific variability of MBW chemical composition	Allows production of SBPs tailored to specific applications	
Product categories	High-performance biosurfactants [22] and composite plastic materials	Plastic materials with better workability and mechanical properties
SPB potential productivity	SBPs produced only from residual fermented MBWs (Section 3)	Higher productivity of SBPs produced from as-collected MBW (Section 3)
Production flexibility of MBW-based biorefinery (Section 1.3 and Section 3)	The adoption of the two models allows the biorefinery modulation of the relative production of biogas and SBPs, depending on the changes in the market demands and prices, as happens for the oil refineries [19]	

## Data Availability

All the available data are reported in the present and in the previous referenced publications.

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
