# Peer review of "Mild Chemical Treatment of Unsorted Urban Food Wastes"

_molecules, 2023, doi:10.3390/molecules28227670_

Round 1

Reviewer 1 Report

Comments and Suggestions for Authors

The topic is of interest for the readers but the text is difficult to follow. It should be more clear and concise.

The state of the art is not clear, authors should define the current gaps and the need of the research to justify the novelty of this work.

The data should be presented in a more clear way. A list of abreviations is needed to facilitate the understanding.

Standard deviation should be given in the data analysis.

The proposed option should be highlighted indicating the main advantages. A comparison table could be used.

Comments on the Quality of English Language

English should be revised

Reviewer 2 Report

Comments and Suggestions for Authors

The objectives are clear in the study of municipal waste and the work is also interesting, with a great abundance of data.

 However, I think that excessive use of abbreviations is made, which makes it difficult to read. They are unusual abbreviations which forces the reader to constantly search for their meaning. For example, is MBW the same as MFW?

For example, table 3 is very difficult to analyze with that presentation.

In summary, the results are interesting, but the writing does not favor understanding due to an excess of acronyms. It seems more like a technical report than a scientific article.

Round 2

Reviewer 1 Report

Comments and Suggestions for Authors

The paper has been improved following the recomendations. A final text editing is needed.